# Co-Powering Solutions to Truck Pollution in South Stockton

**Catherine Garoupa** [1,*], **Nahui Gonzalez Millan** [1], **Bianette Perez** [2], **Taylor Williams** [3] **and Todd Sax** [4,†]

1  Central Valley Air Quality Coalition, 343 East Main Street, Suite 901, Stockton, CA 95202, USA
2  Little Manila Rising, 2154 South San Joaquin Street, Stockton, CA 95202, USA
3  San Joaquin Community Foundation, 6731 Herndon Pl, Stockton, CA 95219, USA
4  California Air Resources Board, 1001 I Street, Sacramento, CA 95814, USA; todd.sax@dtsc.ca.gov
*  Correspondence: catherine@calcleanair.org
†  Todd Sax was affiliated with the California Air Resources Board at the time of authorship; with the California Department of Toxic Substances Control at the time of publication.

**Abstract:** Despite decades of literature and practice with community-engaged research along with advancements in the recognition of environmental injustices, the application of equity-/justice-based and collaborative approaches between government agencies and community-based organizations has been limited. The toxic legacies of environmental racism, redlining, displacement, and segregation combined with the accelerating human-caused climate crisis warrant an increased need for consultation and collaboration between frontline communities and power brokers to markedly improve quality of life and health outcomes in environmental justice neighborhoods. This paper describes the processes and progress to date from a community-led collaboration between local community-based organizations and the Enforcement Division of the California Air Resources Board to assess and address air pollution in South Stockton, particularly from heavy-duty diesel trucks. South Stockton is one of the most polluted neighborhoods in California's San Joaquin Valley, one of the most disparate and polluted regions in the United States. Some of the most significant components integrated into this project thus far include taking an equity-, justice-, and youth-oriented approach to community development that intentionally emphasizes a historical understanding of root causes of social and environmental injustices and provides pathways to workforce development. Including these elements has been essential in building the trust necessary to transform disparate power relations between the state and environmental justice communities, and to put multiple ways of knowing into conversation with each other to co-learn and co-power solutions to air pollution in South Stockton.

**Keywords:** environmental justice; community-based participatory action research; air pollution; San Joaquin Valley

## 1. Introduction and Background

This paper summarizes lessons learned from designing and implementing a transdisciplinary, equity- and justice-based approach to community-based research, action, and enforcement to address air pollution in South Stockton. Despite a robust body of literature and practice with community-engaged research along with advancements in the recognition of environmental injustices in the United States and globally, application of equity-based and collaborative approaches between government agencies and community-based organizations has been lacking. The toxic legacies of environmental racism, redlining, displacement, and segregation in the U.S. combined with the accelerating human-caused climate crisis warrants an increased need for consultation and collaboration between frontline communities and power brokers to markedly improve quality of life and health outcomes in environmental justice neighborhoods.

Community-based participatory action research (CBPAR) provides critical opportunities for co-creation, power sharing, and interdisciplinary learning. A core goal of CBPAR is to blend a variety of expertise and epistemologies or "ways of knowing" (London et al. 2013).

Building on relationships established over the past several years, community-based environmental justice advocates with the Central Valley Air Quality Coalition, Little Manila Rising, and Edge Collaborative are collaborating with the Enforcement Division of the California Air Resources Board (CARB) on a co-designed, community-led effort to address environmental concerns in South Stockton, one of California's most polluted neighborhoods, with severe health impacts such as scoring in the 99th percentile for asthma risk on CalEnviroScreen 4.0 (Office of Environmental Health Hazard Assessment 2021).

*Project and Partners*

Little Manila Rising (LMR) (2022) organizes equitable solutions to the effects of historical marginalization, institutionalized racism, and harmful public policy in South Stockton. LMR offers a wide spectrum of programs that address education, environment, redevelopment, and public health. Their Environmental Justice Program works to reduce environmental hazards like air and water pollution, create sustainable communities by planting trees and vegetative barriers, promote environmental advocacy, and create pathways to green jobs.

Edge Collaborative is a civic incubator that seeds and supports emergent initiatives by deploying capital and providing hands-on support with talent, policy, and coalition-building.

Since 2003, the Central Valley Air Quality Coalition (CVAQ) has raised awareness about pollution issues and their disparate health impacts in the San Joaquin Valley, spearheading policy advocacy to restore clean air to the eight-county region, one of the nation's poorest and most polluted places.

The California Air Resources Board (2020a) adopts regulations designed to reduce air emissions to support the state's air quality and climate goals, and to protect public health. CARB's enforcement program focuses on ensuring that these emission reductions are achieved in practice. CARB staff conduct inspections and leverage technology to identify non-compliance; then, they use legal tools to bring responsible parties into compliance—including penalties that deter future violations.

The collaborative project to co-power solutions to truck pollution in South Stockton emerged from a CVAQ campaign for accountability regarding the Emission Reduction Credit program at the San Joaquin Valley Air District. One outcome of the ongoing effort was a program review that was led by staff of CARB's Enforcement Division. The program review itself took over a year and resulted in findings of mismanagement and corruption that continue to be followed up on (California Air Resources Board 2020b; Earthworks 2018; Central Valley Air Quality Coalition et al. 2023). Through that process, CVAQ and CARB enforcement staff developed a basic level of trust through regular communication, informational workshops, and reciprocal consultation. Advocates and community leaders called out the limitations of positivist, quantitative approaches to air pollution management, exposing their potential for corruption and inherent bias toward a racialized "polluter industrial complex" (Faber 2008). Simultaneously, the selection of Stockton as a priority via the Community Air Protection Program (promulgated by Assembly Bill 617 by C. Garcia; California Legislature 2017) promised opportunities for community-informed enforcement that never bore fruit due to myriad factors. One fundamental flaw in the implementation of AB 617 has been putting local air districts in charge of the process, which replicates existing power dynamics, particularly in the San Joaquin Valley, where the air district is captured by the very industries it is supposed to enforce regulations on (Kennedy 2020). One of the highest priorities repeatedly expressed by community members through the planning process was the commitment to enhanced enforcement. Thus, this project sprung from a foundation of trust and in response to a program that, from advocates' perspectives, did not center community expertise, subverting the methodology by prioritizing incentives for polluting industries over enforcement and community protections. This project created a parallel process demonstrating that community-engaged efforts, when building

from an equity- and justice-oriented CBPAR framework, are productive in both processes and outcomes.

Based on a desire to build durable examples of successful community and agency collaborations, local partners agreed to invite enforcement staff to learn directly from community members about their concerns. After a tour led by advocates to highlight local impacts for CARB enforcement staff in fall 2021, the team of CBOs and CARB's Enforcement Division established group agreements and an initial work plan. Phase 1 of the project, launched in January 2022 and still ongoing, involves assessing and addressing heavy-duty diesel truck traffic using a variety of tools and field observations. Diesel soot is a known carcinogen and poses a serious health risk to those living in close proximity to heavily trafficked roadways, such as Highway 4 and Interstate 5, which are major transportation corridors cutting through South Stockton. Trucks also impact the health, safety, and quality of life for residents living near magnet sources like warehouses, distribution centers, and the port of Stockton through noise pollution, damaged roads, and other issues.

Our group processes have intentionally integrated environmental justice principles such as the intersectionality of social factors such as race, gender, and class (The Combahee River Collective Statement 1977; Collins 2000; Gilmore 2008; Walker and Bulkeley 2006) as well as foregrounding where people live, work, eat, and play (Cole and Foster 2001; Novotny 2000) while confronting racialized capitalism (Bullard et al. 2008; Gilmore 2018; Pulido 2017) and the "polluter industrial complex" that creates unjust conditions (Faber 2008). Recognizing the historical roots and intergenerational impacts of environmental injustice necessitates a decolonizing, community-based participatory action research approach (Denzin and Giardina 2009; Denzin et al. 2008; Kovach 2010; Steinhauer 2002). Key elements include building trust through frequent meetings and discussions led by community advocates, as well as by collaboratively developing working agreements, work plans, and other deliverables. The team is currently collecting data focused on the impacts of heavy-duty diesel trucks in the community through training youth on impacts and interventions like enforcement activities as well as collaborative methods of data collection such as conducting truck counts. Truck counting involves using a standardized form and set days and times to record the number of types of trucks on a particular roadway.

This paper first examines foundational literature on community-based participatory action research (CBPAR), environmental justice (EJ), and youth community development. Next, the community context in South Stockton and the San Joaquin Valley as well as the key players are described. An outline of how research methods and data collection integrate CBPAR and indigenous knowledge provides a framework for understanding the benefits of this approach, particularly related to workforce development for youth and ground truthing enforcement for CARB as an agency dominated by positivist science. The conclusion connects how using an approach rooted in decolonization and CBPAR supports environmental justice principles of intersectionality; engages locally impacted community members, particularly youth; and ground truths the state's enforcement activities to co-learn and co-power solutions to heavy-duty diesel truck pollution in South Stockton.

## 2. Community-Based Participatory Action Research (CBPAR), Environmental Justice, and Youth Community Development

Community-based participatory action research (PAR) and indigenous research methodologies provide instructive frameworks and tools for working with groups that have been traditionally marginalized, like environmental justice communities in the San Joaquin Valley. Spatial analyses of California's San Joaquin Valley have consistently demonstrated that exposure to environmental hazards is concentrated in communities of color and in areas with lower socioeconomic status as well as other demographic indicators such as linguistic isolation (Office of Environmental Health Hazard Assessment 2021; Liévanos 2015; London et al. 2011). While technical debates persist over precisely how to identify environmental justice communities (for example, Mohai and Saha 2015), environmental justice (EJ)

advocates are actively engaged in a variety of strategies to dismantle the processes creating injustice, which is what Pellow (2000) refers to as "environmental inequality formation".

Relational reciprocity, relationship building, accountability, and giving back to the community are key research components that seek to decolonize, heal, and facilitate transformation at the personal and societal levels (Creswell 2013; Denzin 2006; Denzin and Giardina 2009; Denzin et al. 2008; Watson-Gegeo and Watson-Gegeo 2001; Kovach 2010; Sayer 2010; Steinhauer 2002).

Within the environmental justice movement, collaborations between scholars and advocates have generated compelling data documenting the disparate exposure to environmental hazards borne by communities of color and low-income areas that are often facing various other layers of disadvantage. "Toxic Wastes and Race in the United States", published in 1987, was one of the first reports to demonstrate race as a primary factor in the siting of toxic facilities and is a landmark in fully launching the environmental justice movement in America (Bullard et al. 2008; Pulido 2000; Sze and London 2008). Several efforts in California are examples of engaged research that have significantly bolstered advocates' efforts to document and address environmental injustices in various regions of the state (such as Pastor et al. 2007; London et al. 2011). These types of studies document local disparities while demonstrating their unique layering across space and place, point out avenues for actions, and stimulate further refinement of the measurements and methodologies used to delineate environmental justice communities (Sze and London 2008). Building and sustaining trust within the community one researches is supported by reciprocal relationships where researcher(s) and participants engage in mutually respectful exchanges and consider how research can benefit all involved (Denzin and Giardina 2009; Denzin et al. 2008; Kovach 2010; Lashley 2016; Steinhauer 2002). This reciprocal dynamic between researchers and community contributes to a sense of accomplishment while also motivating residents who recognize the significant challenges faced by their local community and have chosen to stay and reinvent their locality (Anguelovski 2013).

Positivist science has also been used to identify environmental justice communities as easy targets for the siting of environmental hazards (Cerrell Associates, Incorporated 1984) and to challenge experiences shared by communities as anecdotal and lacking objectivity and rigor (Cole and Foster 2001; Harrison 2011; Kurtz 2009; Sze 2007), presenting a double-edged sword for communities engaging with research (Hale 2008). Environmental scholars' and advocates' close attention to on-the-ground issues facing frontline communities has also contributed to accusations of practicing a "militant particularism" that is too focused on local circumstances at the expense of understanding broader influences (Harvey 1996; Swyngedouw and Heynen 2003). Yet, local organizers and activist scholars generate many valuable, conceptual insights (Lipsitz 2008) and achieve critical wins essential for building and sustaining social movements.

Given the current primacy of positivist science, particularly embedded in agencies dominated by hard sciences such as the California Air Resources Board, objectivity and rigor are common challenges leveled against community and/or collaboratively generated knowledge (Hale 2008). Yet, giving up the "mantle of objectivity" can result in the practically grounded, relevant results needed to address environmental justice issues locally and globally (Sze and London 2008). To generate research meaningful to movements, recognizing multiple forms of knowledge is crucial, as is reclaiming social scientific methods that some cast as outside the conventions of modern science but that, in fact, can demonstrate strong methodological rigor (Hale 2008). This approach aligns with community-based participatory action research, where the purpose is to bring multiple forms of knowledge into conversation to generate unique insights and outcomes. Furthermore, community-engaged research can help counter the "apartheid" described by Gilmore (1993) under which the academy "privatizes and individualizes what should be collective and public, and explains away collective identities through group individualism" (p. 77). Putting multiple forms of knowledge into conversation can culminate in more durable, holistic, ground-truthed solutions that truly respect and reflect lived experiences of marginalized

peoples. Engaging community as experts can in fact deepen the rigor, relevance, and reach of research (Balazs and Morello-Frosch 2013; London et al. 2013). Co-learning amongst scholars, agency staff, and community members through collective efforts and actions can ensure alignment with the goals of the community (Anguelovski 2013; Eisenhauer et al. 2021; Lashley 2016; London et al. 2018; Minkler and Wallerstein 2012; Powell 2009).

Social change efforts must be situated within their respective institutional arrangements, capacities, and actions (Fantasia 1988). Understanding knowledge and its complex interconnections with group's and individual's multiple identities requires theories of "intersectionality (The Combahee River Collective Statement 1977; Collins 2000)" that "emerged first in the context of political struggles against attempts to prioritize one of people's multiple axes of oppression, a practice that inevitably deprives the others of attention and importance" (Hale 2008, p. 23). While individual motivations, moments of protest, and movement outcomes have received wide coverage in social movement literature, less attention has been paid to the shared culture created (Fantasia 1988) or the processes and mechanisms generating these outcomes (Giugni et al. 1999; Polletta 2002, 2006; Swarts 2008). Epistemologies are inevitably influenced by an individual's social and geographical location. How an environmental injustice is expressed depends on the particulars of place, population, time period, and political context (Gilmore 2008, 2018; Pulido 2000; Sze and London 2008).

Work for environmental justice generally seeks to address both the distribution of exposure to environmental hazards while also seeking a fair process that is inclusive of those impacted, often referred to as distributive and procedural justice (Cole and Foster 2001). Along with the distributive and procedural aspects, Schlosberg (2007) adds recognition as a key third component, as participation in the process does not automatically result in a change in outcomes. Cole and Foster (2001) describe meaningful participation as "substantive dialogue among administrators, experts and affected communities along with the opportunity for affected communities to influence the decision-making process" (p. 16). This approach to participation challenges mainstream environmentalism's reliance on "litigation, legislation, and other pathways to environmental change that exclude so-called non experts from participation" (Harrison 2011, p. 10). Recognition in this context emphasizes historical and contextual factors influencing existing social power dynamics, thereby acknowledging the importance of group identity and reinforcing the necessity for participatory processes that value indigenous knowledge as valid expertise. Building power to transform environmental injustice involves negotiating complicated histories tied to categories such as race, class, and gender in order to cultivate solidarity.

Environmental decision making and the arrangement of physical space reflect societal power dynamics, and racism remains a potent causal factor creating clusters of environmental justice communities (Bullard et al. 2008). The EJ movement and scholarships' exposure of historically-rooted social disparities present important challenges to the mainstream environmental viewpoint of the environment as a pristine wilderness requiring preservation. This privileged approach to nature as another good to consume obscures the disparate impacts borne by low-income communities and communities of color, giving supremacy to "a conception of the environment dislocated from relations of social inequality" (Harrison 2011, p. 10). The EJ movement utilizes a more encompassing vision of the environment that pays attention to where people live, work, eat, and play (Cole and Foster 2001; Novotny 2000). EJ literature and organizing will continue to benefit from efforts that trace the linkages between injustices and sociospatial relations. From a messaging perspective as well as in research and policy, the EJ movement has successfully raised the importance of social location and particularly race and class in perpetuating inequality and disparate exposure to environmental hazards (Walker and Bulkeley 2006). Faber's (2008) work on "capitalizing on environmental injustice" draws attention to a "polluter industrial complex" that puts profits ahead of people and the environment, contributing to uneven economic development while exacerbating existing social ills like racism (Gilmore 2008, 2018). The environmental justice movement has demonstrated that history, context, and multiple

identities commingle to create disadvantage, while community organizing continues to spawn various cultural processes and mobilize vehicles to build and sustain change efforts.

Dominant societal paradigms of environmentalism and the "public good" can replicate structural and historic injustices. Confronting racialized capitalism at times demands seeing the state as "a cite of contestation rather than an ally or neutral force" (Pulido 2017, p. 1). Thus, bridging community-based participatory action research with environmental justice praxis requires a critical approach. Gordon da Cruz describes how applying Critical Race Theory and an overall critical approach to community-engaged research demands "race-conscious analyses, asset based understandings of community, and privileging subaltern experiences" (Gordon da Cruz 2017, p. 1). The sociopolitical context of the San Joaquin Valley and Stockton necessitates an approach that recognizes deeply rooted historical inequities and builds trust that change is possible (London et al. 2013; Gilmore 2018).

Integrating youth into CBPAR further offers transformational opportunities while adding another intersectional layer to consider methodologically. While CBPAR theoretically foregrounds community expertise, engagement exists on a spectrum, as does working with youth, from consultation to collaboration to youth-led; these pathways should be calibrated by a range of factors and can be complementary (Arenstein 1969; Freechild Institute for Youth Engagement 2011; Pei Wu et al. 2005). Mentoring youth is important, recognizing that there are times where positioning young people as leaders is inappropriate and can even be detrimental (Dalzell and Stefansson 2005; London 2007; Sutton 2007). One of the most important assets participatory activities can provide are opportunities for interpersonal and community development that inspire hope (Ginwright et al. 2005; Pei Wu et al. 2005). Non-profits, despite being embedded within a neoliberal, capitalist structure, can develop youth's "political consciousness" and engage them in sociopolitical processes that lead to long-term movement participation (Terriquez 2015). Bearing these theories and reviews of practice in mind, next is an examination of the layers of space, place, and power dynamics this project is nested within.

## 3. Environmental Injustices in South Stockton and the San Joaquin Valley

Stockton sits along the San Joaquin river, adjacent to the Sacramento–San Joaquin delta, a vital ecosystem that is the largest freshwater tidal estuary on the west coast (Selby 2019). Stockton made headlines during the 2009–2010 financial crisis for becoming the first major metropolitan city in the United States to declare bankruptcy. Once the largest population of Filipinos outside the Philippines before the neighborhood was destroyed by the construction of Highway 4, Stockton is one of the nation's most ethnically diverse cities (McPhilips 2020). Highway 4, the "crosstown freeway" was constructed, despite fierce opposition, to facilitate traffic in and out of the Port of Stockton (POS). It destroyed the heart of the Little Manila neighborhood and Barrio del Chivo (Bohulano Mabalon 2013). The POS houses and attracts multiple significant sources of air pollution, from ocean-going vessels; off-road equipment; and adjacent stationary sources such as the biomass plant DTE Stockton, Schuff Steel, and Kinder Morgan, to name just a few (California Air Resources Board 2022). Air toxics come from human-made sources and can cause cancer, birth defects, brain damage, and other serious health impacts. Particle pollution from diesel-powered equipment such as trucks is known to cause cancer, and much of South Stockton is in the 99th percentile for diesel particle pollution exposure (Office of Environmental Health Hazard Assessment 2021). The cumulative health impacts caused by freight and goods movement have been well documented at large ports such as the Port of Oakland and the Los Angeles/Long Beach port complex, particularly for those in close proximity to port operations, rail yards, and/or heavily trafficked roadways (Hricko 2008). These operations and their pollution footprint continue to expand as international trade increases (Hricko 2012).

In 2019, the South Stockton area was selected for the Community Air Protection Program, established by Assembly Bill 617 (C. Garcia, California Legislature 2017), which is meant to take a local, community-based approach to addressing deeply rooted, dangerous

levels of air pollution in some of California's most polluted communities. While the process and program itself have been fraught, numerous other advocacy efforts led by community-based organizations have borne fruit, including through the development of this project.

California's San Joaquin Valley, spanning a broad swath of the state's core, is a region of sociopolitical and environmental contrasts. It is a land of great abundance that yields myriad agricultural products feeding the nation and world, a wealth extracted on the backs of the region's most vulnerable populations that has caused unprecedented, irreversible desecration of the physical landscape. The Valley has seven of the ten most agriculturally productive counties in the U.S. (Economic Research Service 2023), generating more than USD 25 billion in gross annual production (U.S. Environmental Protection Agency 2015) juxtaposed with some of the state's highest rates of food insecurity (Chaparro et al. 2012). As has been well documented, the prominent economic status of the region is built around the exploitation of an inexpensive, and often sociopolitically isolated, immigrant farm labor population (Cole and Foster 2001; Harrison 2006; London et al. 2008). This paradox has led some to describe the region as "poverty amidst prosperity" (Martin and Taylor 1998). The San Joaquin Valley air basin is the most polluted for fine particles and competes with Los Angeles for being the most polluted in terms of ozone (American Lung Association 2022), with significant health and economic consequences. One study showed that attaining clean air standards for ozone and particulate matter 2.5 microns or smaller in the Valley would save the region's economy almost USD 6 billion annually due to reduced deaths and save medical costs such as a decrease in cases of asthma and chronic bronchitis as well as a reduction in emergency room visits (Hall et al. 2008). The overall political culture of the Valley and in Stockton is primarily conservative, often showing up as one of the only "red" Republican voting areas of a predominately liberal state. Values trend toward local control, with sometimes strident opposition to statewide and/or national efforts. This harsh sociopolitical and physical landscape generates daunting dynamics for environmental justice advocates; yet, with caution, they have proceeded to negotiate for and win changes within these systems (Gilmore 2018; O'Connell 2022).

Over the past two decades, addressing environmental justice issues has become a major priority for the state of California. The Enforcement Division at the California Air Resources Board (CARB) responded by increasing inspections in communities, enhancing transparency, and implementing a new Supplemental Environmental Projects program. In 2021, CARB inspected more than 12,000 trucks and diesel equipment—80% of these inspections were in communities identified using CalEnviroScreen as "disadvantaged communities." The type, location, and compliance status of every inspection CARB conducts each year are published in an on-line mapping tool called the CARB Enforcement Data Visualization System, and CARB publishes compliance rates in each program in an Annual Enforcement Report (California Air Resources Board 2021). Program-specific compliance rates do not statistically differ between disadvantaged communities and other communities. Supplemental Environmental Projects or SEPs are environmentally beneficial projects that a responsible party that violated a CARB regulation can choose to implement to satisfy up to 50% of their penalty in an enforcement action. In 2016, CARB implemented a new program that solicits projects that are championed by, benefit, and sometimes fund community organizations. Since that time, more than USD 25 million dollars have funded projects that install high-efficiency air filters in schools, plant trees, and provide asthma care and education services, among others.

Despite these successes, CARB enforcement continued to hear concerns from local community representatives across the state that it was not doing enough. Overall, the California Air Resources Board is dominated by positivist science, while increasingly espousing an equity-based approach. Along with being driven by data generated from air monitoring and computer models, many CARB positions require a Bachelor of Sciences degree. In an effort to improve program effectiveness, the Enforcement Division began partnering with community groups to better understand local needs, and learn how to

meet them. The development of a working relationship with the Central Valley Air Quality Coalition on a previous project provided an opportunity to work together in Stockton and with local partners Little Manila Rising and Edge Collaborative. This collaboration has become a vital conduit for iterative co-learning about the historical basis for environmental injustices; how those injustices continue today; and how to implement equitable, immediate and long-term interventions.

## 4. South Stockton Truck Counting Project

To embrace multiple ways of knowing within a system dominated by positivist science and baked in injustices such as institutionalized racism has required a carefully crafted, deliberative process. Working with environmental justice communities and particular populations like youth requires a justice- and equity-based, historically informed, intersectional approach. With these factors in mind, the project team established the following goals and objectives:

Goals

- Co-learn with community members, students, and CARB enforcement staff about inequitable impacts, community members' priorities for South Stockton, and potential regulatory interventions.
- Ensure data validity and safety of field visits through standard protocols and required training.
- Produce and share information documenting our findings on air pollution sources and the implemented and potential interventions.
- Have fun!

Objectives

1. Assess truck traffic in select locations in South Stockton, including both the types and volume of trucks.
2. Establish baseline truck counts before planned expansions at the Port of Stockton and other magnet sources.
3. Inform pollution protection and reduction measures, and related policies, practices, enforcement, and other actions, including deployment of the newly created Portable Emissions Acquisition System (PEAQS), which assesses emissions/compliance with state regulations. The PEAQS combines emission measurements when a truck drives underneath it and a license plate reader that when cross-referenced against DMV registration data allows CARB to know which specific truck is passing underneath to identify high emitters. This tool is particularly valuable because, from work on motor vehicle emissions, CARB came to understand that the vast majority of emissions are generated by a small minority of vehicles; minimizing emissions requires looking for and repairing these high-emitting vehicles.

### 4.1. Integrating Youth and Justice

Each year, Little Manila Rising has recruited between eight to twelve young adults from Stockton, aged 17 to 24, to participate in the Youth Advocates Environmental Justice Program. When recruiting youth, staff intentionally outreach through Little Manila's deep connections in South Stockton and a combination of school club involvements, community outreach, tabling at key events, and social media campaigns. In many cases, youth advocates who previously participated in LMR's programs have encouraged their friends and families to get involved.

The goal of Little Manila Rising's Environmental Justice Youth Advocates program is to increase awareness and engagement around environmental justice and climate action in South Stockton. A critical program goal is to empower youth to design their own education and be able to take what they learn to engage with their community (Dalzell and Stefansson 2005; Ginwright et al. 2005; London 2007; Minkler and Wallerstein 2012; Pei Wu et al. 2005; Sutton 2007). Little Manila Rising intentionally works with youth to design

their experience and collect post-session workshop surveys to hear their reflections and find ways to continue to elevate their learning experiences. By running youth programs, Little Manila Rising staff have learned that these programs are often the youth's first time learning about the different environmental issues impacting their neighborhoods. Information on environmental justice issues often can be very dense; so, it has been critical to make information digestible and accessible. According to metrics like Arenstein's classic "Ladder of Participation" (1969) and more recent, youth-focused iterations (Dalzell and Stefansson 2005; Freechild Institute for Youth Engagement 2011), youth have primarily been consulted, and at times collaborated with. Those that have participated in follow-up activities such as site visits and truck counts have had increased opportunities to both learn from and inform enforcement activities in action.

A major component of co-learning necessarily involves contextualizing the history of the Little Manila neighborhood, South Stockton, and racialized violence and oppression. Understanding space, place, and time are critical for finding tools and pathways for transformation. This approach aligns with the literature on community-based research and taking a critical, equity- and justice-based approach (Bullard et al. 2008; Faber 2008; Fantasia 1988; Gordon da Cruz 2017; Hale 2008; London 2007; Sutton 2007; Sze and London 2008).

Including the history of South Stockton and of Little Manila Rising as an organization were integral to sharing historical knowledge. Little Manila Rising initially focused on historic preservation and has expanded to numerous issues in response to community needs, and on air pollution and environmental injustice specifically due to the tragic and deeply felt loss of their co-founder to an asthma attack, Dr. Dawn Bohulano Mabalon. This organizational history coupled with the decades of destruction and disinvestment the South Stockton community has experienced resonates with the youth's lived experiences and motivates their engagement.

Social sciences play a key role in understanding the local context of South Stockton. Analysis and investment bereft of the historical underpinnings of the issue often serve to exacerbate those conditions rather than address them. There is a deep history of discrimination, disenfranchisement, and disinvestment underlying the resource inequities and agency inertia experienced today (Pulido 2017). Limited resources should be filtered through a participatory framework where decisions reflect the intention of the impacted community (Anguelovski 2013; Eisenhauer et al. 2021; London et al. 2013).

This project has intentionally taken a participatory approach to community engagement and community development. Little Manila Rising's cohorts of youth build their capacity through trainings and hands-on activities about the social and environmental justice issues affecting South Stockton, through the lens of understanding the historic injustices and disinvestment that has happened to their community. These experiences build youth's political consciousness and contribute to ongoing civic participation and movement building (Terriquez 2015).

Activities

As part of the 2022 and 2023 programs, Central Valley Air Quality Coalition staff provided several trainings with additional optional field observations and activities: Part 1 focuses on the major sources of air pollution in South Stockton and includes breakout discussions and facilitated conversations to connect youth's lived experiences in their neighborhoods with pollution sources. Part 2 hones in on heavy-duty diesel trucks; their impacts on health and quality of life; as well as potential interventions such as enforcement of existing rules and regulations, truck electrification to eliminate tailpipe emissions, and rerouting trucks outside of neighborhoods. As discussed further in the workforce development section, CARB staff also shared their career trajectories and what the day-to-day of their jobs involve. Part 3 involves a hands-on workshop where CARB enforcement staff bring emissions detection equipment such as Automated License Plate Readers and Forward Looking Infrared Cameras, allowing the youth and staff of community-based organizations

the opportunity to understand how air pollution rules and regulations are enforced and to provide feedback on priority places for enforcement and other regulatory actions.

Subsequent to the training series, so far, several youth from the cohort have participated in truck counting and field visits to observe the Portable Emissions Acquisition System, Automated License Plate Readers, and other equipment in action in South Stockton locations selected through collaborative research and mapping. As a result of these initial rounds of data collection, the youth have deepened their knowledge of pollution issues and interventions related to heavy-duty diesel trucks while gaining exposure to potential career pathways. One of the primary points of feedback shared by the youth was that, while growing up, they had experienced the pollution issues discussed but were never taught about root causes or ways to protect themselves and their community. Through the co-learning process, they forge a connection with their past and their present at a critical moment when they are planning their future (Dalzell and Stefansson 2005).

Along with significant strengthening of individual and community capital (Putnam 2004; Walter and Hyde 2012), truck traffic in the initial area of observation has been impacted by CARB staff providing handouts on acceptable truck routes, along with communication with several important stakeholders such as the California Highway Patrol as well as city and county staff about the need for consistent signage and enforcement of routes, particularly because the first site where CARB has conducted their enforcement activities is across from an elementary school, and children's developing lungs are particularly vulnerable to air pollution.

During the field visits, community members and advocates have also learned about spatial patterns such as the road was a previous freeway offramp, has a restaurant with one of the only public bathrooms available, lacked clear signage, along with observing that qualitative data were generally not being captured by California Air Resources Board staff, which was a point of feedback for improved information sharing. Youth participants remarked on the insights they gained by seeing in action what they had learned about conceptually during the workshops.

### 4.2. Workforce Development

The most recent data from the US Census indicate that the annual per capita income in California is USD 38,576; for Sacramento County, it is USD 34,078; and for San Joaquin County, per capita income is USD 30,628 (2023). By exposing students to community development and regulatory enforcement activities through this project, youth gained insights into career pathways and potentially even job opportunities that provide above-average salaries at both the state and local levels. In addition to the wages that accompany these positions, state employees and staff of community-based organizations enjoy a myriad of high-quality benefits, including health insurance, dental insurance, vision insurance, retirement plans, discounts and other financial incentives, and numerous opportunities for paid time off and leave. Employee benefits have a significant impact on cost of living expenses as well as employees' physical and mental health (Sommers et al. 2017).

The table below (Table 1) includes salary ranges for job postings at CARB at the time of this article's writing. The positions included here are the types of positions that youth were exposed to during CARB's visits to Stockton (2023).

**Table 1.** Salary ranges for job postings at CARB.

| California Air Resources Board Job Title | Salary Range |
|---|---|
| *Air Resources Engineer* | *USD 70,188–USD 131,472* |
| *Air Pollution Specialist* | *USD 64,032–USD 123,924* |
| *Air Resources Supervisor* | *USD 123,732–USD 154,860* |
| *Air Resources Technician* | *USD 44,316–USD 55,500* |
| *Automotive Emissions Test Specialist* | *USD 39,648–USD 54,072* |

In addition to the improved process, outputs, and quality of research from the cohort's community-based approach to participatory action research, there are tangible benefits to exposing young people to high-opportunity career pathways that support emissions reduction and exposure mitigation. Those benefits primarily include the development of marketable skills and social capital for accessing work. While workforce development was not initially listed as a project goal, it became a clear need and added benefit once the opportunity to engage directly with Little Manila Rising's youth advocates emerged.

At the first meeting between CARB and the cohort of Stockton youth leaders, the team asked that CARB staff and staff of community-based organizations (CBOs) specifically refer to their educational backgrounds and career journeys, including skills they developed along the way, training pathways, previous places of employment, and so forth. The primary feature of the meeting for youth was to expose them to how regulatory enforcement is implemented in the field and the role of community advocates in community development and policy advocacy. By jointly exposing the cohort to both the responsibilities of CARB and CBO staff's roles as well as the paths by which staff entered those roles, they developed an experiential understanding of high-quality employment opportunities to which they otherwise would likely not have been exposed.

Some of the learning opportunities that these visits offered pertained to important and marketable skills that youth would need to employ in positions such as those of CARB and CBO staff. These include models of co-learning in which many various stakeholders bring different types and levels of expertise and experience to a problem to be solved, and participants learn through the lenses of others to develop a broader, more complete understanding of the project. Participants were also encouraged to ask questions of any of the partners present, including leaders of community-based organizations, institutional leaders (e.g., dean of a local college campus), CARB staff, and each other.

By engaging with many different types of expertise in the room, youth employed a variety of communication skills. Perhaps the most important of these skills was that of critical thinking. In one example while out in the field conducting truck counts and monitoring for criteria pollutants, CARB staff had identified that there were many trucks that were idling next to an elementary school, where they were not permitted to do so. While the enforcement team primarily focused on the quantitative data they needed to gather from the local traffic, it was members of the youth cohort and CBOs that identified that the curb where trucks were idling, previously painted red to indicate that vehicles are not allowed to stop there, had been worn down to gray. Proposing a simple solution of repainting the curb, the youth cohort employed critical thinking skills while demonstrating the value of their perspective in a space that would otherwise not have included them in a traditional enforcement activity.

Beyond the skills required of professionals in these fields, meetings also provided youth the crucial opportunity to develop a network and build social capital with professionals doing the work that the project entails (Putnam 2004; Walter and Hyde 2012). During at least one meeting in which CARB visited Stockton to demonstrate their field equipment for youth, CARB staff indicated that they would offer guidance through the application process for any of the youth participants who were interested in applying to work for the state.

By developing both marketable skills and a social network of enforcement professionals, youth participating in the program likely increase their likelihood of employment into the types of jobs (or indeed, the specific jobs) that support the enforcement of mobile emissions regulations. In doing so, they are (1) extending the benefits of a more whole process of addressing local air quality issues by bringing learnings from the model into future iterations of regulatory enforcement and (2) increasing their likelihood of enjoying the economic benefits of high-quality employment.

*4.3. Ground Truthing Enforcement*

Partnership with the California Air Resources Board's Enforcement Division has enhanced opportunities to inform and engage community members in data collection and

enforcement activities by providing access to cutting-edge emissions detection equipment and activities such as roadside inspections. These activities have already resulted in improved communication and capacity building among key stakeholders, including CARB enforcement staff. During the collaboration, CARB's enforcement work in Stockton continued, now informed by a deeper understanding of historical factors driving local injustices as well as community perspectives on issues and priorities. CARB staff deployed technology to screen vehicles, conducted inspections, and pursued enforcement when necessary. At the same time, the collaboration provided unique new opportunities. CARB staff participated in educational events where staff shared their career paths and how enforcement technology works. CARB staff collected additional information during inspections that helped answer questions raised by the collaborative about why trucks were taking certain routes and which fleets might be targeted for incentive programs to reduce emissions.

As a component of this collaborative project, CARB staff collected information to better understand the impact of truck routes in the community, including the impacts of trucking traffic on a local elementary school. After a review of existing information about areas of concern gleaned from the Community Air Protection Program planning process, this project ground-truthed data through hands-on, interactive mapping exercises with the youth, advocates, and other community members to determine priority locations for deploying equipment and executing enforcement. Throughout the process, CARB staff gained a greater appreciation and deeper insights into the value of local partnerships to support not only CARB objectives but community priorities as well. In reflecting on the project thus far, CARB staff have shared appreciation for the fact that the community development process is as important, if not more important, than the outcomes. Once data collection and analysis conclude, the team will co-write a report describing the community's concerns, compliance of emissions sources with state and local requirements, the root causes of the community concerns, and steps taken to reduce community impacts, with a focus on vulnerable populations and heavily impacted neighborhoods.

## 5. Conclusions

Working with environmental justice communities and populations such as youth requires a critical, equity- and justice-based approach that encourages historical understanding of root causes, creates a safe space for building political consciousness and movement building, and offers pathways for developing social and community capital as well as access to career pathways (Putnam 2004; Terriquez 2015; Walter and Hyde 2012). Recognizing the legacies of redlining and other forms of institutionalized racism and oppression is vital to meeting environmental justice communities and youth participants where they are, particularly for agents of the state to build trust in the midst of decades of disinvestment and discrimination (Bullard et al. 2008; Cole and Foster 2001; Gilmore 2008, 2018; Kurtz 2009; London 2007; Sze and London 2008; Pulido 2017; Sutton 2007). Utilizing a community-based participatory action research approach allows for community expertise to be put in conversation with the technical, scientific approaches of state enforcement staff, embracing multiple ways of knowing and integrating critical social scientific approaches that help ground positivist science in the lived realities of environmental justice communities (Anguelovski 2013; Eisenhauer et al. 2021; London et al. 2013; Powell 2009; Sutton 2007). Developing and sustaining this group dynamic with our transdisciplinary team relied on several years of preexisting working relationships along with carefully curated processes led by community advocates that included clear meeting agreements and goals as well as frequent meetings to prepare, implement, and debrief ongoing activities. The South Stockton project seeks to build on existing strengths and capacities while developing new skills and innovating solutions to challenging, deeply rooted air pollution and environmental injustices.

Through this collaborative project in South Stockton, community groups and youth participants have gained knowledge of the California Air Resources Board's (CARB's) authority, programs, enforcement activities, and career pathways; further, through the data

collection process, they have gained insights into the nature of and remedies related to heavy-duty diesel truck traffic. CARB has gained insight into how to move beyond routine enforcement work to be a more supportive and progressive partner with communities working to proactively address environmental injustices. CARB staff have learned that while enforcement work is helpful to reduce emissions, it is not sufficient to address long-standing issues in the community, and that the act of working together can support local organizers who are helping to build a greater community. Taking a community-based approach has also helped CARB staff to better understand how focusing on local concerns can both enhance enforcement efforts and allow CARB to play a more supportive role as a helpful and positive resource for people in communities that have historically been disenfranchised or discriminated against by the government.

This project demonstrates that while the state is a critical site of contestation (Pulido 2017), justice-centered CBPAR led by community advocates working with people within entrenched institutions can carve out pathways for innovative and transformative collaboration. The engagement of a well-resourced and powerful agency such as CARB that is, by its nature, dominated by positivist science calls for a critical, equity- and justice-based approach to co-learning and co-powering solutions. All project participants learned to integrate multiple forms of knowledge through collaborative processes while foregrounding community expertise. These components are essential for integrating youth from environmental justice communities into community development while providing them tangible tools and pathways to decolonize, heal, and facilitate transformation at personal and societal levels (Creswell 2013; Denzin 2006; Denzin and Giardina 2009; Denzin et al. 2008; Gegeo and Watson-Gegeo 2001; Ginwright et al. 2005; Kovach 2010; Pei Wu et al. 2005; Sayer 2010; Steinhauer 2002).

**Author Contributions:** Conceptualization, C.G.; methodology, C.G. and N.G.M.; software, C.G. and N.G.M.; validation, C.G., N.G.M., B.P., T.W. and T.S.; formal analysis, C.G., N.G.M., B.P., T.W. and T.S.; investigation, C.G., N.G.M., B.P. and T.S.; resources, C.G. and N.G.M.; data curation, C.G. and N.G.M.; writing—original draft preparation, C.G., N.G.M., B.P., T.W. and T.S.; writing—review and editing, C.G.; visualization, C.G. and T.W.; supervision, C.G.; project administration, C.G. and T.S. All authors have read and agreed to the published version of the manuscript.

**Funding:** This research received no external funding.

**Institutional Review Board Statement:** The Institutional Review Board of California State University, Stanislaus approved the protocol for this research (2122-059) for studies involving humans on 2 February 2022 and renewed 26 June 2023. The study was conducted in accordance with the Declaration of Helsinki.

**Informed Consent Statement:** Informed consent from human subjects was not required for this project because all observed behavior happened in public places and personally identifiable information was not documented.

**Data Availability Statement:** No data is publicly available at this time due to resource and capacity limitations, though it is a long term goal of the project to share information online.

**Conflicts of Interest:** The authors declare no conflict of interest.

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
