# Peer review of "Co-Powering Solutions to Truck Pollution in South Stockton"

_socsci, doi:10.3390/socsci12080440_

Round 1

Reviewer 1 Report

Title: Co-powering Solutions to Truck Pollution in South Stockton

This paper promises to make important contributions to the literature. Collaboration between regulatory agencies and community organizations seeking to address the climate crisis by integrating “local knowledge” into decision making processes is a very promising and well-intentioned strategy for advancing environmental justice. However, as the authors establish well in their CBPAR section, the challenges to meaningful participation and integrations of community knowledge and perspectives into regulatory enforcement are substantial. Engagement in south Stockton provides a valuable case study to help understand these challenges given the area’s extreme environmental injustice and entrenched inequalities. In addition, the community and government agency partners involved have a rich and valuable history in advocating for more equitable processes for “environmental justice” communities by engaging in the hard and fraught work of forging community-government partnership to address disparate air pollution exposure. The contribution of this article could be enhanced with some revisions to sync the various sections and themes together to make the paper more of a cohesive whole, which includes linking some of the important insights from the introduction and CBPAR sections, through the results and into the conclusion.

It would be helpful to have a clear statement of the specific organization of the paper at the end of the introduction. In particular, it will be helpful to set reader expectations that there are three main “results” sections (or that the review is divided into these sections) which I believe are: a review of the project’s CBPAR processes and some benefits of this process, workforce development of youth, and ground truthing enforcement. These sections all make contributions to the paper’s main themes, but they come across now as a bit disjointed and not part of a cohesive whole. I feel setting up reader expectations at the end of the introduction will provide clarity on how the three parts are connected to each other and the main goals of the paper. Then it would be helpful for the authors to connect more strongly each section of the paper with the intro and background literature review in the CBPAR section.

The CBPAR section is one of the best and most up to date review of the relevant issues I have read recently. The section introducing the case of south Stockton and the San Joaquin Valley was also excellent.

The section on the overview of the project (South Stockton Truck Counting Project and Approach) was strong as well, and it sounds like a very interesting and engaging effort. It would be helpful if some of the language and background from the CBPAR section could become visible here, such as: make clear which of the approaches in Stockton reflect or sync with the pretty established CBPAR literature? This section also seems to be doing two things that might be worth separating into different subsections. First, it established the processes used. This section seems more like background or “methods”. To that end, it would be helpful to clearly state the “data” used to identify the benefits or impacts of this process/program. This might be based on participant observations, or perhaps other methods of collecting insights such as groups discussion, reflections, or interviews? Second, this section describes some of the “outcomes” or new insights generated by this process, which might fit in a separate section. That is, this second part seems to draw from observations to make the case of the value generated by this process.

The sections on Workforce Development and on Groundtruthing Enforcement provide valuable insights. It would be helpful if the background section of the paper fleshed out what the literature says about these two benefits of engagement and CBPAR, and then in these sections can more directly make connections with how this program affirms and extends previous insights.

The conclusion connects many of the themes of the paper well and starts to pull it all together. It needs to also connect with how this program and the insights provided relate to previous studies and advance out understanding of the challenges and benefits of community-agency collaboration. Also, it would be helpful to have a description of the study limitations as well as priorities for future research.

Some minor notes:

It would be helpful to know the ages of the youth (teens, young adults), roughly how many, and more details on how were they recruited (and at a general level what other aligned program they were participating in).

Although I prefer a term such as “environmental justice neighborhoods” over a designation such as “disadvantaged communities”, it would be helpful to have some direct but general description of what constitutes an “environmental justice” community (without trying to get into specific metrics). On page 2, what are “environmental justice” principles are being referenced (perhaps a reference will clarify for the reader)?

One sentence of additional details on what is the PEAQS would be helpful. Sounds innovative.

On page 9 in relation to CARB, it uses first person (we, our) which the text should stay in third person.

In summary, I believe after the paper’s major insights are fleshed out a bit more and aligned more with the academic and practice literature, I am confident it will make an important contribution.

Reviewer 2 Report

Your manuscript presents a unique CBPR partnership in an important EJ community and region of California.  It would make an important contribution to the field.  See my comments in the PDF. 

Round 2

Reviewer 1 Report

Overall, the authors have done a nice job of addressing my concerns, making the paper more “cohesive”, and connecting the findings back to the literature in the conclusion. I recommend publication after the authors address some minor issues.

On page 2 line 67, “the corrupt” can be deleted. These words are not needed to make the main points.

On page 2, line 83, the phrase “an agency captured by industry” is unclear and perhaps not needed. It also implies that this was the only or main reason the effort did not bear fruit when in reality there may have been several reasons. (The Kennedy citation listed seems more generally about the composition of regulatory boards, not specifically this CAPP). If the reference to “industry” is kept then perhaps more general phrasing such as “given concerns that industry was overrepresented in the process” would work better. On the page line 87 the word “failed” could be removed because the main point would still be intact (or make clear that it was viewed as a “failure” by community organizations).

On page 3 line 125 this phrase is floating and incomplete: “setting the project’s methodological grounding.”

On page 6, line 273, the phrase “of intergenerational and youth led” can be deleted.

Page 9 line 418: “consulated” should be “consulted”

Page 10, line 470: This should be make two words: “bringemissions”

I look forward to the publication of this important paper, and to citing it in my future work.

Overall this paper is clearly written with high quality English.

Reviewer 2 Report

The substantive and thoughtful revisions have resulted in a much improved article. Reorganization of the paper clarifies and elevates the key arguments in the paper. Adding recent research also makes this a much improved draft.  A close proofread would catch typos and check for journal conventions for use of hyphens (e.g. "place-based",  "community-based", etc. ) but these are minor.  I look forward to seeing this article published!
